# Detection and Parameter Estimation Analysis of Binary Shift Keying Signals in High Noise Environments [note 1]

**DOI:** 10.3390/s22093203

**Published:** 2022-04-21

**Authors:** Van Minh Duong, Jiri Vesely, Petr Hubacek, Premysl Janu, Nhat Giang Phan

**Affiliations:** 1Department of Communication Technologies, Electronic Warfare and Radiolocation, Faculty of Military Technology, University of Defence, 662 10 Brno, Czech Republic; minhktqs2008@gmail.com (V.M.D.); petr.hubacek@unob.cz (P.H.); premysl.janu@unob.cz (P.J.); 2Department of Electronic Warfare, Faculty of Radio-Electronic Engineering, LE QUY DON Technical University, 236 Hoang Quoc Viet, Bac Tu Liem, Hanoi 10000, Vietnam; pngiang20000@gmail.com

**Keywords:** barker code, binary phase shift keying signal, cross-correlation function

## Abstract

In this paper, a new method for detecting and estimating the parameters of a binary phase shift keying (BPSK) signal, based on a cross-correlation function, is proposed. The proposed method consists of two stages. The first stage is used to detect or estimate a signal carrier frequency, and the second stage is used to estimate its pulse width or symbol rate. Firstly, the proposed method is investigated by use of a simulated BPSK signal in the form of Barker Codes 7, 11, and 13 in the MATLAB environment. Based on the simulation results, the functionality of this method is verified using a real-time BPSK signal generated by an E8267C generator. This is described in the second part of this paper. The experimental test results confirm that the proposed method is able to detect and estimate the parameters of all BPSK signals with SNR≥−21 dB.

## 1. Introduction

In passive surveillance systems (PSSs), the BPSK signal is a typical radar signal with a low probability of intercept (LPI), and it has been designed for many modern radar systems [1,2]. Moreover, these signals are transmitted in a high-noise environment. The interception of LPI signals has been a topic of investigation for over a decade. To detect and estimate the parameters of BPSK signals, the Fourier analysis method using Fast Fourier Transform (FFT) was used as a basic tool [3]. From this basic tool, more complex methods have evolved, such as the Short-Time Fourier Transform, in order to estimate the signal parameters over time [4,5]. Later, more effective techniques have also been designed, such as Time-Frequency Distributions, in order to identify the LPI radar signals. These techniques include the Ville–Wigner Distribution, Cyclo-Stationary, and Wavelet Transform [6,7,8,9,10]. These techniques are effective to detect and estimate the parameters of LPI signals with an SNR≥0 dB. In recent years, new techniques have been developed for analyzing BPSK signals, based on Deep Learning (DL) or Artificial Intelligence (AI), such as the Convolution Neural Network (CNN) and Deep Convolution Neural Network [11,12]. These techniques can recognize BPSK signals with an SNR≥−6 dB. One disadvantage of these methods is that their accuracy depends on the database or the total number of samples—the fewer the samples, the lower the accuracy.

Subsequently, a new technique was used to estimate the code rate of the BPSK signals, based on the Nyquist Folding Receiver (NYFR), as presented in [13] or based on a duffing oscillator [14]. The SNR required to achieve a nearly perfect probability of the correct code-rate estimation ($P_{ce}\geq 90\%$) is SNR≥−16 dB. Another method used to detect BPSK signals is called the Parameter-Adjusted Bistable Stochastic Resonance Model, based on a scale change in [15]. This method produces the best results (probability of detection: $P_{d}\geq 90\%$) for input BPSK signals with an SNR≥−18 dB. It shows that all the above-mentioned techniques are capable of detecting and estimating the parameters of BPSK signals in a high-white-noise environment, but they require the database of BPSK signals.

To overcome the previously mentioned drawbacks, a new technique for detecting and estimating BPSK signal parameters with an SNR≤−20 dB (see Figure 1), based on the cross-correlation function, is presented in this paper and in our previous paper [16,17]. This method consists of two stages: the first stage focuses on detecting BPSK signals or estimating their carrier frequency $f_{c}$; the second stage is used to estimate their pulse width τ or symbol rate. Firstly, the proposed method was investigated, using simulated BPSK signals, in the MATLAB environment to predict the lowest SNR, at which the proposed method was still capable of detecting and estimating the BPSK signal parameters ($P_{d}\geq 90\%$). Later, the functionality of the proposed method was verified by real-time BPSK signals generated by an E8267C PSG signal vector generator.

A theoretical description of the proposed technique is presented in Section 2. The accuracy of this method was investigated in the MATLAB environment by different simulated BPSK signals, which are the most commonly used in radar systems, to predict the lowest SNR, at which the proposed method was still capable of detecting and estimating the BPSK signal parameters (90% detection probability and correct estimation) in Section 3. The functionality of this method was verified by real-time BPSK signals in Section 4. The main conclusions are summarized in Section 5.

## 2. The Proposed Technique

In radar signal processing [18], the cross-correlation function is used to measure the similarity of two signals as a function of time in relation to one another. This technique has been used in pattern recognition and single particle analysis. Therefore, the cross-correlation function between two signals, $x\left(t\right)$ and $y\left(t\right)$, in the time domain is defined by Equation (1):(1)$$R_{xy}\left(\delta \right)=\int_{-\infty }^{\infty }x^{*}\left(t\right)y\left(t+\delta \right)dt,$$
where $R_{xy}\left(\delta \right)$ is the cross-correlation function between the two signals, $x^{*}\left(t\right)$ is the complex conjugate of signal $x\left(t\right)$, $y\left(t+\delta \right)$ is the second signal at the time t+δ, and δ is the time delay. On the other hand, the cross-correlation function can be expressed by (2).
(2)$$R_{xy}\left(\delta \right)=F^{-1}\left\{X^{*}\left(\omega \right)\times Y\left(\omega \right)\right\}$$
where $X\left(\omega \right)$ and $Y\left(\omega \right)$ are the spectra of the signal $x\left(t\right)$ and $y\left(t\right)$, $X^{*}\left(\omega \right)$ is the complex conjugate of $X\left(\omega \right)$, and $F^{-1}$ is the inverse of Fourier transform [19].

Figure 2a shows the simulation result of the cross-correlation function (CCF) between two BPSK signals generated with the same carrier frequency ($f_{1}=f_{2}=50\;\mathrm{MHz}$, blue line), and with different frequencies ($f_{1}=50\;\mathrm{MHz};\;f_{2}=20\;\mathrm{MHz}$, red line), at the pulse widths ($\tau_{1}=\tau_{2}=7\;\mu s$) and an SNR=−20 dB. It clearly shows that when two signals have the same parameters (carrier frequencies), their CCFs have the highest value at the time delay δ=0 μs.

For the same case, Figure 2b shows the plot of the CCFs between two BPSK signals generated with the same pulse widths (τ1=τ2=7 μs, blue line) and with different pulse widths (τ1=7 μs; τ2=14 μs, red line) at a carrier frequency of f1=f2=50 MHz. This figure shows that the maximum CCF between the two signals, having the same parameters, is always higher, Rxxδ=39.47 dB, than the CCFs between the two signals having different pulse widths, even if they have the same carrier frequency Rxyδ=37.53 dB. From these simulation results, a new technique for detecting and estimating the parameters of BPSK signals, based on the cross-correlation function, was proposed in our previous papers. The proposed method technique was divided into two stages. The first stage is used for detecting the BPSK signal or estimating its carrier frequency fc, and the second stage is used for estimating its pulse width or symbol rate τ. A block diagram of this technique is shown in Figure 3. This figure clearly shows that two sets of reference BPSK signals are used. The signals in the first set of reference signals were generated with the same pulse width and with varying carrier frequencies. The signals in the second set were generated with the same carrier frequency, estimated in the first stage of the proposed method, and with different pulse widths. The steps of the proposed method are the following (Algorithm 1).
**Algorithm****1** Steps of the proposed method **Input parameters:**fref,τref are the parameters of the reference BPSK signals.**Output parameters:**
fc, τ are the parameters of the received BPSK signals.**Step 1:** Calculating the spectra of the received Xω and first set of reference BPSK signals Xrefω using FFT.**Step 2:** Calculating the CCF between the received signal and the first set of reference BPSK signals R1δ using (2).**Step 3:** Finding out the maximum R1δ as a function of the carrier frequency fref1 of the reference BPSK signals or g1f.**Step 4:** Finding out the maximum g1f to estimate fc.**Step 5:** Calculating the spectrum of the second set of reference BPSK signals using FFT.**Step 6:** Calculating the CCF between the received signal and the second set of reference BPSK signals R2δ using (2).**Step 7:** Finding out the maximum R2δ as a function of the pulse width τref2 of the reference BPSK signals or g2τ.**Step 8:** Finding out the maximum g2τ to estimate τ and calculating Pd
and Pce.

Figure 4 shows the plot of the peak CCFs between the received signal and the first set of reference BPSK signals as a function of their carrier frequency $g_{1}\left(f\right)$. In this example, the carrier frequency $f_{c}$ of the received BPSK signal is set out of the carrier frequency range of the reference signals ($f_{c}\notin f_{ref}$, see Table 1). This figure shows that maximum $g_{1}\left(f\right)$ is always at the highest ($f_{c}=99\;\mathrm{MHz}$ with $g_{1}\left(f\right)=35.60\;\mathrm{dB}$, see Figure 4a) or at the lowest carrier frequencies of the reference BPSK signal ($f_{c}=101\;\mathrm{MHz}$ with $g_{1}\left(f\right)=35.18\;\mathrm{dB}$, see Figure 4b). It means that the frequency bandwidth of the reference signals is not correctly set. From these simulation results, the frequency bandwidth $f_{c}\in f_{ref_{1}}$ needs to be adjusted (see Table 2).

In order to detect a signal or estimate its carrier frequency $f_{c}$, the index of the reference BPSK signal, which provides the highest CCF with the received signal, or the maximum $g_{1}\left(f\right)$ are required. The maximum CCF as a function of the carrier frequency $f_{ref_{1}}$ of the reference signals is shown in Figure 5a. This figure shows that the maximum $g_{1}\left(f\right)$ is $g_{1\max}\left(f\right)=29.02\;\mathrm{dB}$ at f=100 MHz. It also shows that the unknown BPSK signal is close to the reference signal and its carrier frequency is estimated as $f_{c}=100\;\mathrm{MHz}$. In other words, the unknown BPSK signal was detected, or its carrier frequency was estimated during the first stage of this method.

The next step in this work is to generate the second set of reference BPSK signals with the carrier frequency estimated during the first stage of this technique, and with varying pulse widths ranging from 0.1 to 20 μs. The following step is to calculate the second CCF between the received signal and the second set of reference signals. As previously mentioned in relation to the estimation of the carrier frequency $f_{c}$ of the BPSK signal, the reference signal index or the maximum $g_{2}\left(\tau \right)$ are required. The maximum CCF between the received signal and the second set of reference signals $R_{2}\left(\delta \right)$ as a function of the pulse width $\tau_{ref_{2}}$ or a function of $g_{2}\left(\tau \right)$ is plotted in Figure 5b. This figure shows that the highest $g_{2}\left(\tau \right)$ is $g_{2\max}\left(\tau \right)=32.32\;\mathrm{dB}$ at τ=7 μs. It means that the pulse width of the received BPSK signal was estimated during the second stage of this technique. Table 3 lists the estimated parameters of the received BPSK signal with Barker Code 7 at an SNR=−20 dB, acquired by running the system for 200 loops.

The main advantage of the presented technique is its ability to simultaneously measure of the parameters of the pulse and the detection of pulse under noise, which lead to its ability to detect signals in lower SNR environments.

In the next step of this paper, the accuracy of the proposed technique was investigated with different simulated BPSK signals, such as Barker Codes 7, 11, and 13. It was simulated in the MATLAB environment to predict the lowest SNR, at which this technique still achieved 90% of the detection probability and the correct pulse width estimations for all the BPSK signals.

## 3. Simulation Results

A theoretical description of the proposed technique for detecting and estimating the BPSK signal parameters was presented in our paper [16,17]. In this section, the proposed technique accuracy was investigated in the MATLAB environment, with different simulated BPSK signals (different type of Barker Codes), by running the system for 500 loops at an SNR ranging from −35 to −20 dB. All the tests were performed under the condition that the parameters of the received BPSK signals were within the observed parameters of the reference signals ($f_{c}\in f_{ref_{1}};\;\tau \in \tau_{ref_{2}}$).

The detection probability $P_{d}$ for all the simulated BPSK signals as a function of the SNR is shown in Figure 6a. It is clear that the proposed technique achieved the highest detection probability for the simulated BPSK signal with Barker Code 13 ($P_{d}=97.24\%$ at an SNR=−30 dB, red line), followed by the BPSK signal with Barker Code 11 ($P_{d}=94.60\%$ at an SNR=−30 dB, blue line), and the lowest detection probability was for Barker Code 7 ($P_{d}=92.20\%$ at an SNR=−30 dB, black line). Ultimately, the lowest SNR was an SNR=−30 dB in order to reach a 90% detection probability for all the signals.

The same principle was applied to the estimation of the $f_{c}$ of the received signal. The probability of the correct pulse width estimation of all the BPSK signals as a function of the SNR is presented in Figure 6b. This figure shows that this method achieved the best result for the BPSK signal with Barker Code 13 ($P_{ce}=92.35\%$ at an SNR=−26 dB, red line), followed by Barker Code 11 ($P_{ce}=92.92\%$ at an SNR=−24 dB, blue line), and the worst result was for Barker Code 7 ($P_{ce}=93.55\%$ at an SNR=−21 dB, blue line) in terms of the SNRs. The lowest SNR, at which this technique was still able to reach 90 % of the correct pulse width estimation probability for all the test BPSK signals ($P_{ce}\geq 90\%$), was SNR≥−21 dB. In the next section of this paper, based on the simulation results, the functionality of this technique was verified by real-time BPSK signals generated using an E8267C vector signal generator.

## 4. Experimental Test Results

In the previous section, the accuracy of the proposed technique was investigated by simulated BPSK signals in the MATLAB environment. Based on these simulation results, the proposed technique was applied to the real-time BPSK signals to verify its functionality. In the first part of this section, the proposed method was verified by the BPSK signal with Barker Code 7 at an SNR=−21 dB; and in the second part of this section, the method was verified at an SNR=−24 dB for Barker Code 11. Finally, the functionality of this technique was verified with Barker Code 13 at an SNR=−26 dB. All the experiments were performed under the condition that the parameters of the received BPSK signals were within the observed parameters of the reference BPSK signals ($f_{c}\in f_{ref_{1}};\;\tau \in \tau_{ref_{2}}$).

### 4.1. Test Setup

A block diagram of the experimental test setup is shown in Figure 7. This figure shows that an E8267C PSG vector signal generator was used for generating BPSK signals in the form of Barker Codes, which operated in the frequency range from 250 kHz to 20 GHz. The PSG E8267C generator generated basic radar signals, such as single- or multicomponent AM, FM, CW, or QAM signals, implemented in its hardware. Additionally, this generator generated special radar signals, for which the parameters were set from external sources, such as MATLAB, through LAN communication. The generated BPSK signals were saved in a csv format, which was then used to verify the functionality of the proposed technique. In addition, a DSA 814 spectrum analyzer and RTO 1044 oscilloscope were used as amplitude detectors, or a classic method was used based on FFT to analyze the generated BPSK signals. The generated BPSK signal in the form of Barker Code 7 without noise and its spectrum are shown in Figure 8.

### 4.2. Test with a BPSK Signal in the Form of Barker Code 7

Firstly, the functionality of this method was verified with the generated BPSK signal using Barker Code 7 at an SNR=−21 dB. The generated BPSK signal at an SNR=−21 dB and its spectrum are plotted in Figure 9. It is clear that the classic method based on FFT, or the spectral analyzer and oscilloscope were not able to detect and estimate the generated signal parameters.

The maximum CCF between the generated signal and the first set of reference BPSK signals $R_{1}\left(\delta \right)$ as a function of the carrier frequency $f_{ref_{1}}$ of the reference signals ($g_{1}\left(f\right)$) is presented in Figure 10a. This figure clearly shows that the maximum $g_{1}\left(f\right)$ is $g_{1\max}\left(f\right)=-2.86\;\mathrm{dB}$ for f=100 MHz. It is clear that the unknown BPSK signal was detected by the first stage of the proposed technique and its carrier frequency is $f_{c}=100\;\mathrm{MHz}$. The same applied to the estimation of $f_{c}$ for the generated signal. Figure 10b shows the plot of the maximum CCF between the generated signal and the second set of reference signals $R_{2}\left(\delta \right)$ as a function of their pulse width $\tau_{ref_{2}}$, or function $g_{2}\left(\tau \right)$. It is clear that the maximum $g_{2}\left(\tau \right)$ is $g_{2\max}\left(\tau \right)=-5.13\;\mathrm{dB}$ at τ=7.18 μs. The estimated parameters of the BPSK signal generated at an SNR=−21 dB are listed in Table 4. The experimental test results confirm that the proposed technique is able to detect and estimate the parameters of BPSK signals with an SNR≥−21 dB. In the next part of this section, the functionality of the proposed technique is investigated with the BPSK signal in the form of Barker Code 11 at an SNR=−24 dB.

### 4.3. Test with a BPSK Signal in the Form of Barker Code 11

In this part, the functionality of the proposed technique is verified with a BPSK signal generated with Barker Code 11 at an SNR=−24 dB. The generated BPSK signal and its spectrum are presented in Figure 11. It shows that the FFT was not capable of detecting and estimating the carrier frequency, nor the pulse width of the generated signal.

Figure 12a shows that the plot of the maximum CCF between the generated signal and the first set of reference signals $R_{1}\left(\delta \right)$ depended on their carrier frequency $f_{ref_{1}}$ or function $g_{1}\left(f\right)$. It is clear that the maximum $g_{1}\left(f\right)$ is $g_{1\max}\left(f\right)=-4.20\;\mathrm{dB}$ at f=100 MHz. It means that the carrier frequency of the generated BPSK signal was estimated by the first stage of this technique as $f_{c}=100\;\mathrm{MHz}$. The same applied to the estimation of the carrier frequency, and the maximum CCF between the generated signal and the second set of reference signals $R_{2}\left(\delta \right)$ as a function of their pulse width $\tau_{ref_{2}}$ ($g_{2}\left(\tau \right)$) is shown in Figure 12b. This figure shows that the highest $g_{2}\left(\tau \right)$ is $g_{2\max}\left(\tau \right)=-6.17\;\mathrm{dB}$ at τ=11 μs. The estimated parameters of the generated BPSK signal in the form of Barker Code 11 at an SNR=−24 dB are listed in Table 3. The experimental test results confirm that the proposed technique is able to detect and estimate the parameters of the BPSK signals with Barker Code 11 with an SNR≥−24 dB. In the next section, the functionality of this technique is verified with BPSK signal generated in the form of Barker Code 13 at an SNR=−26 dB.

### 4.4. Test with a BPSK Signal in the Form of Barker Code 13

In the last part of this section, the functionality of the proposed technique is verified with a BPSK signal in the form of Barker Code 13 generated at an SNR=−26 dB. The generated BPSK signal and its spectrum are shown in Figure 13. It is clear that the classic method based on FFT, or DSA 814 spectral analyzer and RTO 1044 oscilloscope were not capable of detecting and estimating the parameters of the generated BPSK signal at an SNR=−26 dB.

Figure 14a shows the plot of the maximum CCF between the generated signal and the first set of reference signals $R_{1}\left(\delta \right)$ as a function of their carrier frequency $f_{ref_{1}}$ (function $g_{1}\left(f\right)$). It is clear that the maximum $g_{1}\left(f\right)$ is $g_{1\max}\left(f\right)=-6.17\;\mathrm{dB}$ at f=100 MHz. It means that the carrier frequency ($f_{c}=100\;\mathrm{MHz}$) of the generated BPSK signal was estimated during the first stage of the proposed technique. The same applied to the estimation of the carrier frequency. Figure 14b shows the maximum CCF between the generated signal and the second set of reference signals $R_{2}\left(\delta \right)$ as a function of their pulse width $\tau_{ref_{2}}$ (function $g_{2}\left(\tau \right)$). It is clear that the highest $g_{2}\left(\tau \right)$ is $g_{2\max}\left(\tau \right)=-6.17\;\mathrm{dB}$ at τ=13 μs. The estimated parameters of the generated BPSK signal are listed in Table 4. The experimental test results confirm that the proposed technique is able to detect and estimate the parameters of BPSK signals in the form of Barker Code 13 with an SNR≥−26 dB.

Overall, the experimental test results show that the proposed technique is able to detect and estimate the parameters (carrier frequency and pulse width) of the BPSK signal in the forms of Barker Codes 7, 11, and 13 in a high-white-Gaussian-noise environment.

The last experiment in this section investigates the accuracy of our method, in comparison to other methods, for detecting and estimating BPSK signal parameters using Barker Code 7 in a high-noise environment. The detection probability of each method is shown in Figure 15. This figure shows that the proposed technique achieved better results than the other methods analyzing BPSK signals in a high-white-noise environment. At an SNR=−18 dB, our technique provided the best detection probability ($P_{d}=100\%$, red line), followed by a parameter-adjusted algorithm ($P_{d}=94.49\%$, black line) in [15], and the lowest result was for NYFR ($P_{d}=63.95\%$, blue line) in [13]. The summarized simulation and experimental test results show that the proposed technique is more effective than the existing methods for analyzing BPSK signals in [13,15]. Moreover, the proposed method is simple to implement into any hardware of an electronic warfare receiver. In the future work, the accuracy of the proposed technique will be examined with multicomponent BPSK signals with Barker Codes 7, 11, and 13 in a high-white-noise and interference environment using CW signals.

## 5. Conclusions

The main goal of this article was to increase the range of passive ELINT systems on a radar with a BPSK intra-pulse modulation, where Barker Code lengths 7, 11, and 13 were the most frequent. The main concern was to find out if these pulses are evenly irradiated by the sidelobes of a radar with a high SLS antenna when the parameters of the pulse (CF, PW, and code) are unknown.

A new technique for detecting and estimating the parameters of a BPSK signal with Barker Codes 7, 11, and 13 in a high-white-noise environment, based on a cross-correlation function, was presented and verified in this paper. The simulation results show that this technique is capable of detecting and estimating the parameters of all simulated signals at an SNR≥−21 dB.

In the second part of this paper, based on the simulation results, the functionality of the proposed technique was verified with real-time BPSK signals, generated by an E8267C PSG vector signal generator. The experimental test results show that this technique is able to detect and estimate the parameters of the BPSK signal for Barker Code 7 at an SNR≥−21 dB, for Barker Code 11 at an SNR≥−24 dB, and for Barker Code 13 at an SNR≥−26 dB.

Moreover, the proposed technique was more effective than the existing methods currently used to detect and estimate the parameters of BPSK signals (their carrier frequency and pulse width). The proposed technique requires the lowest SNR≥−30 dB for detecting the signal, while the existing methods require an SNR≥−18 dB in [15] and an SNR≥−16 dB in [13].

In future work, the proposed technique will be applied to multicomponent BPSK signals using different types of Barker Codes (such as Barker Codes 7, 11, and 13) or different types of phase-coded signals (such as Polyphase Barker Codes and Frank Codes) in a high-noise and -interference environment using CW signals.

## Figures and Tables

**Figure 1 sensors-22-03203-f001:**
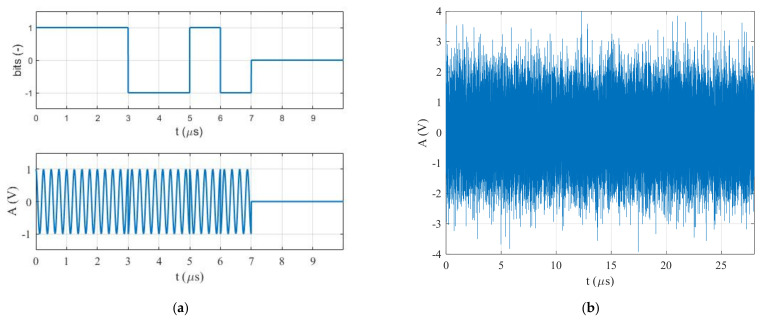
The BPSK signal in the form of Barker Code 7: (**a**) without noise, and (**b**) SNR=−20 dB.

**Figure 2 sensors-22-03203-f002:**
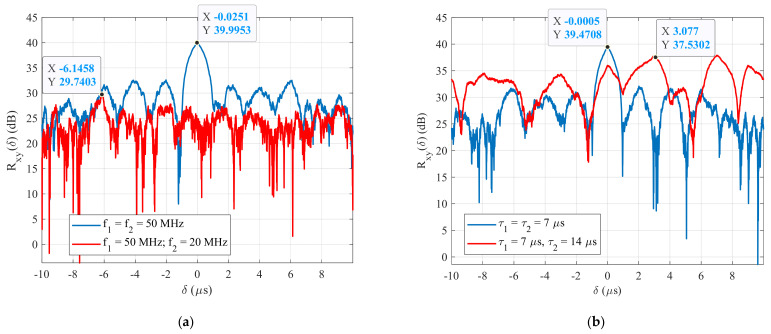
The CCF between two BPSK signals at an SNR=−20 dB; (**a**) carrier frequency, and (**b**) pulse width.

**Figure 3 sensors-22-03203-f003:**
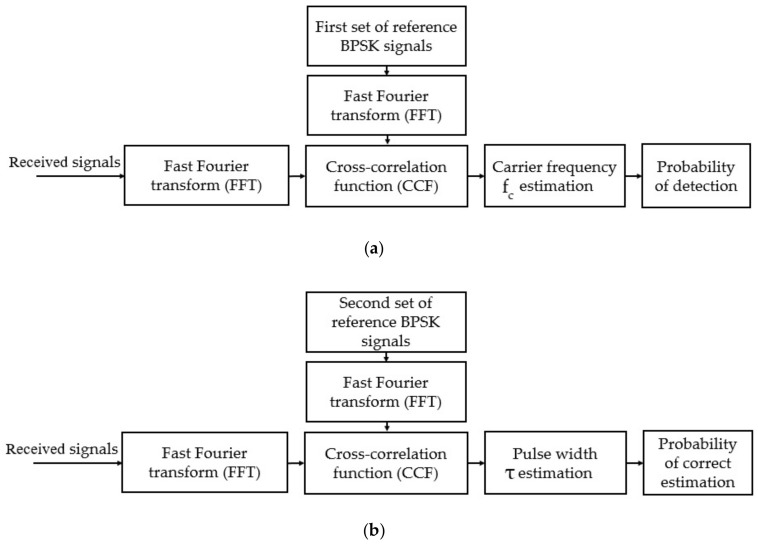
The proposed method: (**a**) detection process, and (**b**) pulse width estimation.

**Figure 4 sensors-22-03203-f004:**
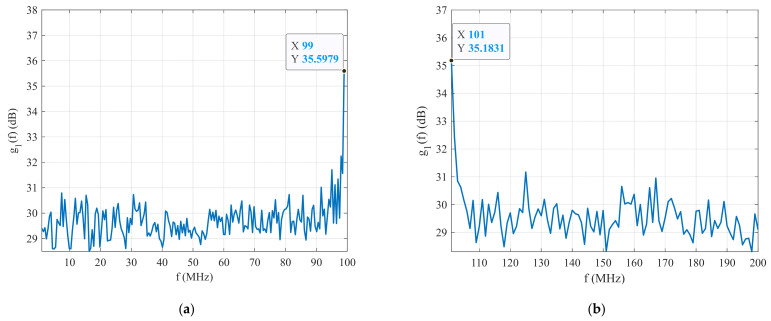
The CCF between the received and first set of reference signals: (**a**) $f_{c}>f_{ref_{1}}$, and (**b**) $f_{c}<f_{ref_{1}}$.

**Figure 5 sensors-22-03203-f005:**
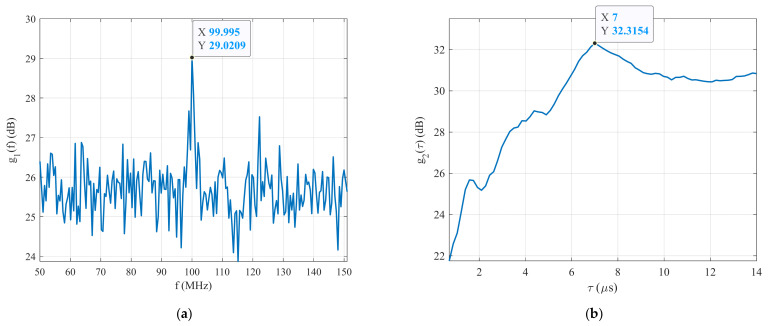
The simulation results: (**a**) carrier frequency estimation, and (**b**) pulse width estimation.

**Figure 6 sensors-22-03203-f006:**
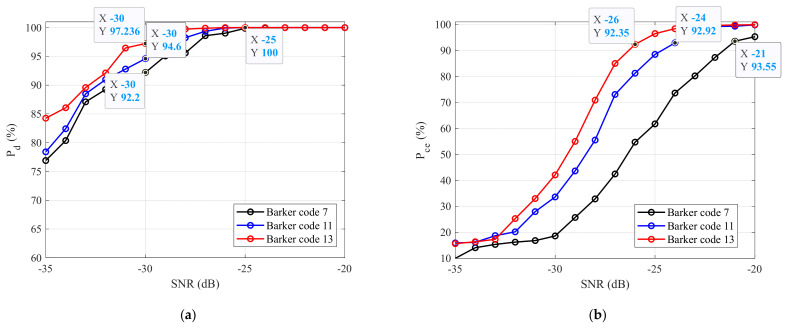
The simulation results: (**a**) probability of detection, and (**b**) probability of pulse width estimation.

**Figure 7 sensors-22-03203-f007:**
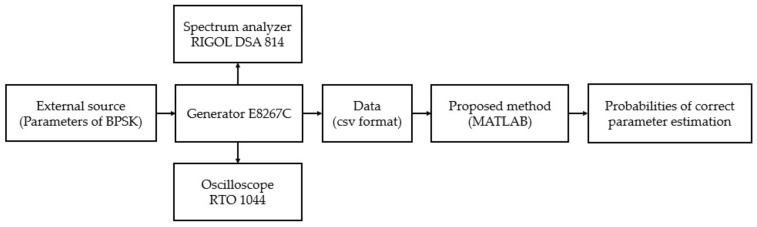
Block diagram of the experimental test setup for verifying the functionality of the proposed technique.

**Figure 8 sensors-22-03203-f008:**
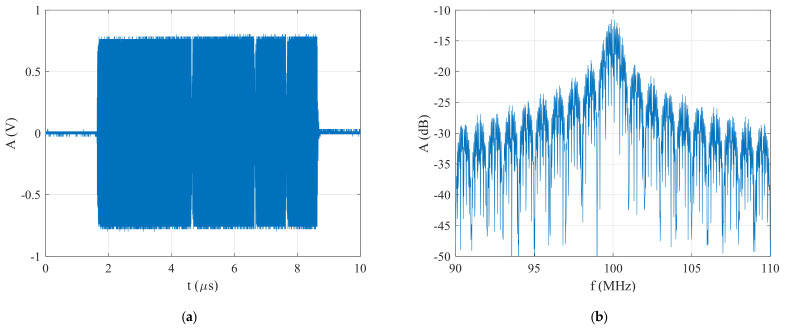
Generated BPSK signal in the form of Barker Code 7 without noise: (**a**) time domain, and (**b**) spectrum.

**Figure 9 sensors-22-03203-f009:**
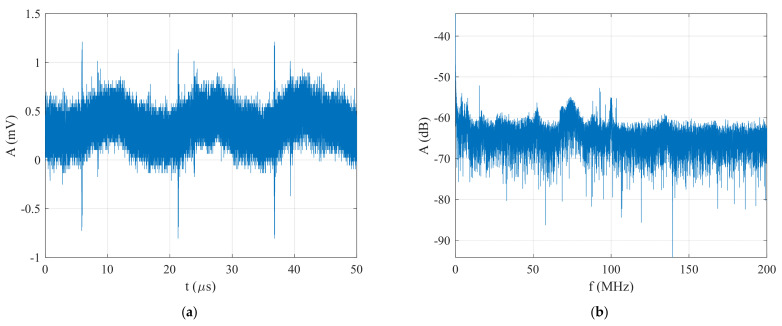
The generated BPSK signal in the form of Barker Code 7 at an SNR=−21 dB: (**a**) time domain, and (**b**) spectrum.

**Figure 10 sensors-22-03203-f010:**
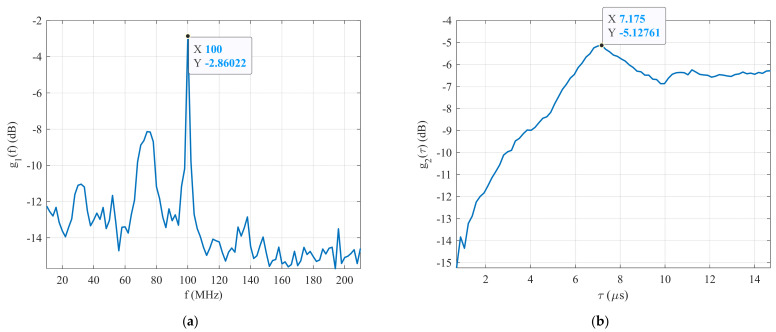
The simulation results: (**a**) detection process, and (**b**) pulse width estimation.

**Figure 11 sensors-22-03203-f011:**
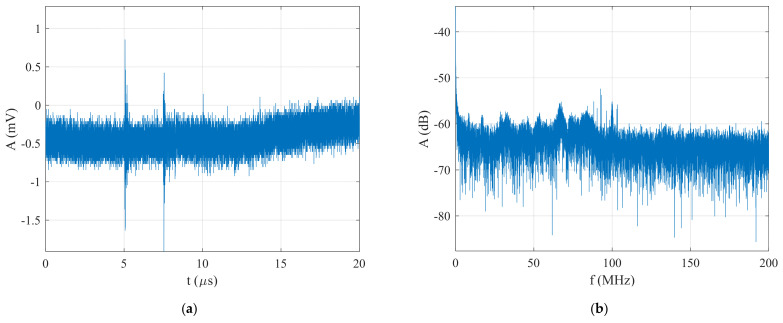
The generated BPSK signal in the form of Barker Code 11 at an SNR=−24 dB: (**a**) time domain, and (**b**) spectrum.

**Figure 12 sensors-22-03203-f012:**
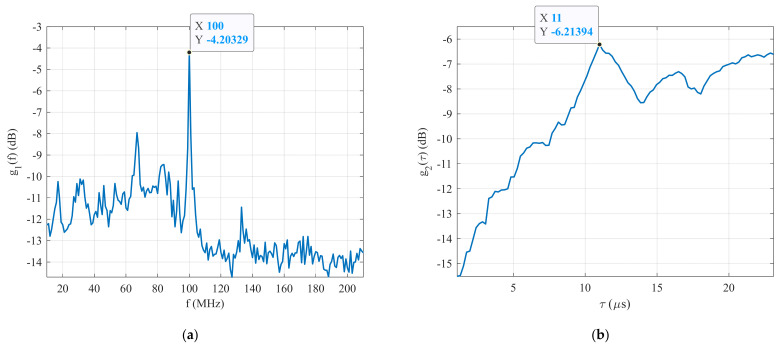
The simulation results: (**a**) detection process, and (**b**) pulse width estimation.

**Figure 13 sensors-22-03203-f013:**
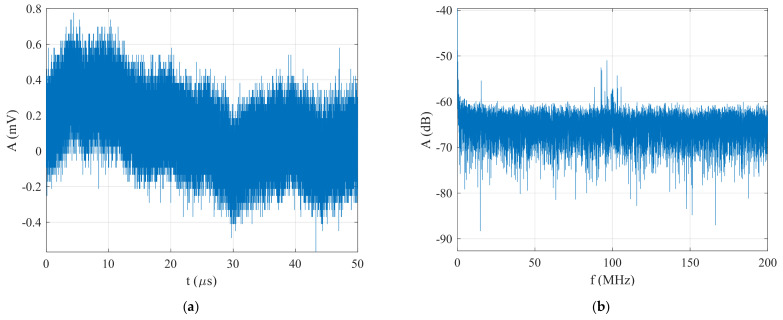
The generated BPSK signal in the form of Barker Code 13 at an SNR=−26 dB: (**a**) time domain, and (**b**) spectrum.

**Figure 14 sensors-22-03203-f014:**
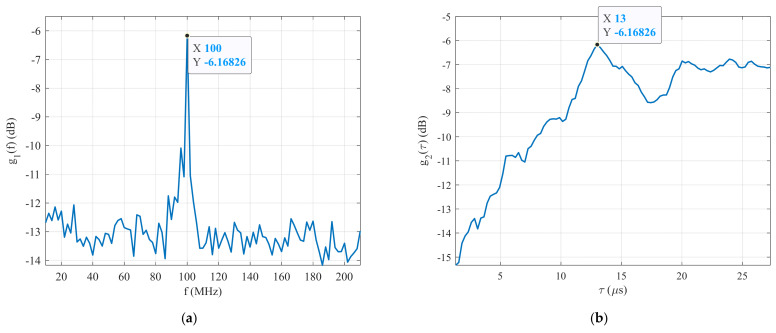
The simulation results: (**a**) detection process, and (**b**) pulse width estimation.

**Figure 15 sensors-22-03203-f015:**
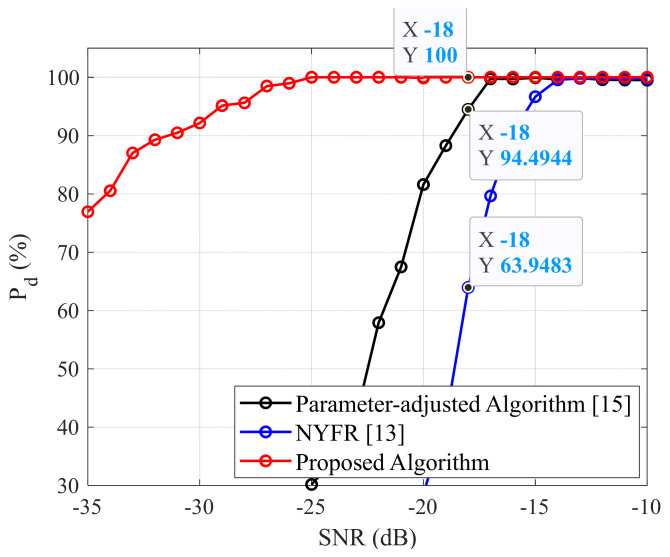
Detection probability of each method depending on the SNR.

**Table 1 sensors-22-03203-t001:** Signal parameters for the first simulation.

Signals	Parameters	Values
Received signal	$f_{c}\;\left(MHz\right)$	100
	$SNR\;\left(dB\right)$	−20
Reference BPSK signals	$f_{ref}\;\left(MHz\right)$	1÷99 or 101÷200

**Table 2 sensors-22-03203-t002:** Parameters of the simulated signals.

Signals	Parameters	Values
Received signal	$f_{c}\;\left(MHz\right)$	100
	$\tau \;\left(\mu s\right)$	7, 14, 13
	$SNR\;\left(dB\right)$	−20
	Barker Code	7, 11, 13
First set of reference BPSK signals	$f_{ref_{1}}\left(MHz\right)$	50÷150
	$\tau_{ref_{1}}\left(\mu s\right)$	14, 22, 26
	Barker Code	7, 11, 13
	Number of reference signals	100
Second set of reference BPSK signals	$f_{ref_{2}}\left(MHz\right)$	100
	$\tau_{ref_{2}}\left(\mu s\right)$	0÷20
	Barker Code	7, 11, 13
	Number of reference signals	100

**Table 3 sensors-22-03203-t003:** Estimated parameters of the received BPSK signal at an SNR=−20 dB.

Parameters	Simulation Value	Estimated Value	Relative Error (%)	RMSE (-)
$f_{c}\;\left(MHz\right)$	100	100	0.0	0.0
$\tau \;\left(\mu s\right)$	7	7.65	9.21	7.29

**Table 4 sensors-22-03203-t004:** Estimated parameters of the generated BPSK signals in the form of Barker Codes 7, 11, and 13.

Signals	Parameters	Simulation Value	Estimated Value	Accuracy (%)
Barker Code 7	$f_{c}\;\left(MHz\right)$	100	100	100
	$\tau \;\left(\mu s\right)$	7	7.18	97.5
Barker Code 11	$f_{c}\;\left(MHz\right)$	100	100	100
	$\tau \;\left(\mu s\right)$	11	11	100
Barker Code 13	$f_{c}\;\left(MHz\right)$	100	100	100
	$\tau \;\left(\mu s\right)$	13	13	100

## Data Availability

Not applicable.

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
