# Peer review of "Detection and Parameter Estimation Analysis of Binary Shift Keying Signals in High Noise Environmentsâ€"

_sensors, 2022, doi:10.3390/s22093203_

Round 1
Reviewer 1 Report
This paper proposes a novel method to detect and estimate the parameters of BPSK signal based on the cross-correlation function. The proposed method consists of two stages. The first stage is used to detect or estimate the carrier frequency of signal and the second is used to estimate its pulse width or symbol rate. The proposed method is investigated by simulated BPSK signal in the
MATLAB environment and by real-time BPSK signal generated using generator E8267C. Some issues are listed in the following.
1. The estimated values of the carrier frequency and the pulse width of signal are all obtained at the hightest point of CCF. But for the received signal with strong radom noise, the hightest value of CCF may be at approximate the same carrier frequency or pulse width.
2. How to evaluate the performance of the proposed method in the real enviroment with multipath interference?
3. Only white Gaussian noise is taken into account in the received signal, but the different channels will influence the received signal. Will this method still work perfectly in different channels?
4. What the length of sampling data in the calculation of CCF?
5. Can the theoretical explanation be provided for the accuracy of the proposed method?
6. How the probability of detection and probability of the pulse width estimation are defined in Figure 6?
7. For the high carrier frequency(for example, exceed 10GHz), the direct RF sampling digital receiver is costly. Can the proposed method been applied to the IF signal with lower frequency?
Author Response
Dear Reviewer
We thank you for your review, comments, and recommendation. We incorporated them into the letter.
2. How to evaluate the performance of the proposed method in the real environment with multipath interference?
When reflected signals levels will be comparable the maximum CCF will be smeared. If the reflected signal level will be higher than the direct signal, the evaluated signal will be the reflected signal and the direct signal will be suppressed. It can lead to an error in TOA which cal lead to an error in a location in the TDOA method – this is a practical problem with comparison to not detecting weak impulses or detecting but located in wrong position.
3. Only white Gaussian noise is taken into account in the received signal, but the different channels will influence the received signal, will this method still work perfectly in different channels?
Not. the main problem in ESM or ELINT system is to use the system gain of intrapulse modulation. The main limitation is the thermal antenna noise at receiver frond end where Gaussian noise is prevailing so this is the reason why different types types of noise was not tested.
4. What the length of sampling data in the calculation of CCF:
we used the sampling frequency fv = 2 GHz, the length of the generated data is:
T = 50 (us)
5. Can the theoretical explanation be provided for the accuracy of the proposed method
Not exactly, because this method is not optimal, but practical. There are possible enhancing of SNR level of detection.
6. How the probability of detection and probability of the pulse width estimation are defined in Figure 6.
The probability of detection is the probability the algorithm is able to find ou the pulse is present or not if the pulse is present for given SNR level
The probability of pulse width estimation is the probability the algorithm is able to estimate the appropriate pulse width. The main limitation of the algorithm is not the accuracy of measured parameter but the ability to do it in a given SNR level environment
7. For the high carrier frequency ( for example, exceed 10 GHz) the direct RF sampling digital receiver is costly. Can the proposed method been applied to the IF signal with lower frequency?
In practice, the method is evaluate on intermediate frequency (IF = 750 MHz, BW = 250 MHz) or on zero IF frequency on 12-bit - IQ data flow (modernized receiver)
Thank you
have a nice day
Reviewer 2 Report
The paper proposes a novel method for detecting and estimating Binary Phase Shift Keying (BPSK) signals in high-noise environments.
I would suggest a rephrasing of the paper’s title. The authors could consider a title like “Detection and Parameter Estimation Analysis of Binary Shift Keying Signals in High Noise Environments”.
Regarding authors’ statement in lines 10-12, my impression is that the paper is rather an application of the method described in ref. [15] (regarding BPSK signals) and an extension of ref. [16].
In eq. (1), the integration should be with respect to t.
In eq. (2), F-1 is the inverse of Fourier Transform, not FFT (FFT is the algorithm for evaluating Fourier Transform).
In lines 61-67, section 3 should be explicitly mentioned (regarding simulation results that are already referred to).
In lines 10 and 53-54, reference to [15-16] should be rephrased (the “my previous paper” could be omitted).
The paper needs extensive editing regarding the use of English.
All in all, the paper does include novel material (presented in an adequate and scientifically sound manner) and as such is publishable (subject to the comments made above).
Author Response
Dear Reviewer
We thank you for your review, comments, and recommendation. We incorporated them into the letter.
1. I would suggest a rephrasing of the paper’s title. The authors could consider a title like “Detection and Parameter Estimation Analysis of Binary Shift Keying Signals in High Noise Environments”.
Yes. the title of the paper was changed.
2. In eq. (1), the integration should be with respect to t.
Yes. it was corrected in the paper
3. In eq. (2), F-1 is the inverse of Fourier Transform, not FFT (FFT is the algorithm for evaluating Fourier Transform).
Yes. it was corrected in the paper
4. In lines 61-67, section 3 should be explicitly mentioned (regarding simulation results that are already referred to).
Yes. it was corrected in the paper
5. In lines 10 and 53-54, reference to [15-16] should be rephrased (the “my previous paper” could be omitted).
Yes. it was corrected in the paper
6. The paper needs extensive editing regarding the use of English.
Yes.
thank you
Reviewer 3 Report
This manuscript ( ID: sensors-1644683; Title :The Binary Phase Shift Keying Signal Detection and Parameter Estimation Analysis in the High Noise Environment) investigates two phase shift keying signal detection and parameter estimation. The method describes the carrier frequency of the detection or estimation signal and is used to estimate its pulse width or symbol rate. In view of the content and arrangement of this paper, the following aspects need to be improved.
1. In the title, it is suggested to remove the definite article "The".
2. The introduction is not sufficient. It is suggested to investigate and quote the research results related to binary phase shift keying signals in the last three years.
3. The citation order of references should be in ascending order. Please adjust and modify the citation of the literature that appears for the first time in the introduction or the citation of the literature in the literature list.
4. In subsection 2, line 73, the expression of X (T) and Y (T) is not accurate enough. Here, x, y and t should be in italics. Check the writing of other variables at the same time.
5. Fig. 4, how is the frequency interval determined and what is the basis?
6. For the Figure 6(a), please make sure the blue solid line is marked with "Barker code 11" instead of "Baker code 11". Also, check the icons of other diagrams.
7. The conclusion part needs to be compressed and refined, especially the first paragraph of the conclusion.
Author Response
Dear Reviewer
We thank you for your review, comments, and recommendation. We incorporated them into the letter.
1. In the title, it is suggested to remove the definite article "The".
Yes. The title of the paper was changed.
2. The introduction is not sufficient. It is suggested to investigate and quote the research results related to binary phase shift keying signals in the last three years.
I mentioned references from the basic methods to advanced methods in the terms of SNR values. all references are in the last three years
3. The citation order of references should be in ascending order. Please adjust and modify the citation of the literature that appears for the first time in the introduction or the citation of the literature in the literature list.
Yes. I have done it in the paper
4. In subsection 2, line 73, the expression of X (T) and Y (T) is not accurate enough. Here, x, y and t should be in italics. Check the writing of other variables at the same time.
Yes. I corrected it in the paper
5. Fig. 4, how is the frequency interval determined and what is the basis?
The main limitation of the frequency interval is computation complexity. The main source of information is previous measured frequency interval where signal was measured with high SNR (due to irradiating ESM receiver by main lobe)
6. For the Figure 6(a), please make sure the blue solid line is marked with "Barker code 11" instead of "Baker code 11". Also, check the icons of other diagrams.
I have done it in this paper
7. The conclusion part needs to be compressed and refined, especially the first paragraph of the conclusion.
yes. I changed it
Thank you
have a nice day
Round 2
Reviewer 1 Report
This paper has been revised accordingly.
Author Response
Dear Reviewer
We thank you for your review, comments, and recommendation
have a nice day
Reviewer 3 Report
The author made some modifications to the manuscript ( ID: sensors-1644683), but some details were not corrected:
1. Although many references within 3 years are cited. However, in the introduction, the document code should start from serial number 1 and increase to the last document in ascending order, which also determines the document number in the document list。
2.In Section 2, the innovation of the presented technology is insufficient.
3.The significant decimal places of data need to be consistent.
4. The conclusion is lengthy, which shows that the content of the result discussion is insufficient.
Author Response
Dear Reviewer
We thank you for your review, comments, and recommendation. We incorporated them into the letter.
1. Although many references within 3 years are cited. However, in the introduction, the document code should start from serial number 1 and increase to the last document in ascending order, which also determines the document number in the document list.
I corrected them on this paper.
2. In Section 2, the innovation of the presented technology is insufficient.
I agree and I rewrote section 2 in this paper.
3. The significant decimal places of data need to be consistent.
Yes. I agree and I corrected them on the paper.
4. The conclusion is lengthy, which shows that the content of the result discussion is insufficient.
I rewrote the conclusion.
thank you for your information!
have a nice day